# Feeding Dairy Goats Dehydrated Orange Pulp Improves Cheese Antioxidant Content

**DOI:** 10.3390/vetsci11040171

**Published:** 2024-04-11

**Authors:** José Luis Guzmán, Luis Ángel Zarazaga, Antonio Ignacio Martín-García, Manuel Delgado-Pertíñez

**Affiliations:** 1Departamento de Ciencias Agroforestales, Escuela Técnica Superior de Ingeniería, Universidad de Huelva, “Campus de Excelencia Internacional Agroalimentario, ceiA3”, Campus de la Rábida, 21819 Palos de la Frontera, Spain; guzman@uhu.es (J.L.G.); zarazaga@uhu.es (L.Á.Z.); 2Estación Experimental del Zaidín (CSIC), Profesor Albareda 1, 18008 Granada, Spain; ignacio.martin@eez.csic.es; 3Departamento de Agronomía, Escuela Técnica Superior de Ingeniería Agronómica, Universidad de Sevilla, Ctra. Utrera km 1, 41013 Sevilla, Spain

**Keywords:** alternative feedstuffs, rennet substitute, goat cheese, orange by-product, Payoya breed, fatty acid profile, antioxidant capacity, phenolic compounds, fat-soluble vitamins

## Abstract

**Simple Summary:**

Agroindustrial by-products serve as an alternative source of livestock feed, significantly contributing to the sustainability of livestock systems. A very interesting by-product is dehydrated orange pulp (DOP), generated by the industrial production of orange juice, Spain being the largest producer of it in Europe. The DOP could be used as food for ruminants, replacing cereals. We have shown that DOP could replace up to 80% of the cereals in diets without detrimental effects on milk production during complete lactation and the growth of the kids. Furthermore, the milk quality improved from the perspective of human health. As a continuation of that trial, and given the current consumer interest in the healthy properties of animal products, the present study examined the replacing cereal with DOP in the diet of dairy goats, and focused on the antioxidant compounds and fatty acid contents of artisanal cheeses made from raw milk. The vitamin E content, total phenols, and total antioxidant activity in the cheeses increased as the amount of dehydrated orange pulp in the diet rose. In summary, the use of dehydrated orange pulp contributes to both sustainable livestock practices and the quality of goat cheese.

**Abstract:**

Agroindustrial by-products constitute an alternative source of feed livestock, and their use contributes to the sustainability of livestock systems and the circular bioeconomy. The effects of replacing cereal (0%, 40%, and 80%) with dehydrated orange pulp (DOP) in the diet of goats on the antioxidant and fatty acid (FA) contents of cheeses were evaluated. For a more suitable understanding of the role of coagulant enzymes in establishing the properties of the cheese, the effect of milk-clotting with animal and vegetable rennet was also analysed. The rennet did not substantially affect the FA or the antioxidant compounds, and the use of DOP did not affect the FA contents. However, the α-tocopherol levels, total phenolic compounds (TPC), and total antioxidant capacity (TAC) in cheeses increased as the percentage of DOP replacing cereals increased. Moreover, the high correlation obtained between the TAC and the TPC (r = 0.73) and α-tocopherol (r = 0.62) contents indicated the important role played by these compounds in improving the antioxidant capacity of the cheese. In conclusion, DOP is a suitable alternative to cereals in the diet of goats and improves the antioxidant status of the cheese produced.

## 1. Introduction

Agroindustrial by-products are an alternative food source for livestock, and their use contributes significantly to the sustainability of livestock systems and the circular bioeconomy [1,2]. Within the category of possible by-products, Spain is the largest producer of dehydrated orange pulp (DOP, dry residue of orange peels, pulp, and seeds) in Europe (2.8 million tonnes in 2022) [3]. DOP is generated by the industrial production of orange juice, which can represent up to 15% of the original raw material. This by-product can be used as food for ruminants, replacing cereals [4] and, additionally, a source of bioactive compounds such as phenols and vitamin E [5].

Few papers have investigated the replacement of cereals with DOP in goats on the milk yield and fatty acid (FA) and antioxidant properties [6,7], and no reports are available in cheeses. Recently, the effect of the partial replacement of cereals by DOP pellets (40% and 80% replacement) in the feed of lactating goats has been investigated, primarily on production and the growth of the kids [8,9] and secondarily on the quality of the milk [2,8]. The DOP by-product could replace up to 80% of the cereals in diets without detrimental effects on milk production during a complete lactation and the growth of the lactating kids. Furthermore, the quality of the milk improved from the approach of human health: the contents of vitamin E, total phenols, and antioxidant activity increased. Nowadays, foods with natural antioxidants are more popular and sought after for their beneficial repercussions on human and animal health [10,11]. Thus, the improvement in quality due to the use of DOP could improve the economic valorisation of the products and contribute to the sustainability of small ruminant livestock systems.

In Europe, especially in Spain, the milk from dairy goats is mainly destined to produce cheese [12]. Therefore, to better support the applicability of using DOP as an alternative and sustainable source of dairy goat food that increases the valorisation of the products, the quality of the cheese must be analysed. The effects of partial substitution of cereals with DOP on the physicochemical and organoleptic properties of the cheese have been studied [13]. Considering the concerns related to animal rennet (i.e., high cost, restricted natural provisions, religious arguments) [14], Guzmán et al. [13] have also investigated the use of rennet substitute, in particular vegetable enzymes, in cheese manufacture. More specifically, the cardoon flower (*Cynara cardunculus*) was evaluated because it has traditionally been used to produce some Mediterranean artisanal ewe and goat cheeses [13,15]. The results showed that neither the diet nor the type of coagulant substantially modified the quality aspects analysed in the study.

As a continuation of that trial and given the current consumer interest in the healthy properties of products, the present study examined the replacing cereal with DOP in the diet of dairy goats on the antioxidant compounds and FA contents of cheese. In addition, because to the best of our knowledge, no trials dealing with the effect of rennet substitutes on specific antioxidants, such as phenolic compounds and fat-soluble vitamins, have been reported, their effects on the parameters of milk-clotting preparations with animal and vegetable rennet have been analysed for a more suitable understanding of the role of coagulant enzymes in establishing the properties of the cheese.

## 2. Materials and Methods

### 2.1. Animals, Experimental Diets, and Cheese Manufacture and Sampling

The animals, their experimental diets, and cheese manufacturing and sampling procedures have been described in Guzmán et al. [13]. Briefly, 44 Payoya dairy goats were divided into three dietary groups: control (CD, *n* = 14), in which the diet consisted of alfalfa hay plus commercial concentrate; DOP40 (*n* = 16) based on CD, but with 40% of the cereals in the concentrate replaced by DOP; and DOP80 (*n* = 14), based on CD, with 80% of the cereals in the concentrate replaced by DOP. In the fifth month of lactation, the animals were offered the experimental diets, which were adapted for this lactation month. The chemical compositions of the diets are described in Table 1. For more details on diet and animal management, see also Delgado-Pertíñez et al. [2].

At the beginning of the fifth month of lactation, approximately 20 kg of bulk-tank milk per batch for cheese manufacture was collected from each dietary group and transported refrigerated to a cheese factory. Two additional batches were processed under the same conditions in two consecutive days. Half of each batch was clotted with animal rennet and the other half was clotted using vegetable coagulant (*Cynara cardunculus* L.), considering the manufacturer’s instructions. The cheeses were made without a starter culture and with unpasteurised milk, following the manufacturing conditions described in Guzmán et al. [13].

A total of 18 cheeses were made, with three replicate samples for each diet group and rennet type. After ripening, the cheeses were transported refrigerated to the laboratory, cut into pieces, and frozen at −20 °C, or at −80 °C in the case of the samples for vitamins, for later analysis.

In addition, representative milk samples (50 mL aliquots from each animal) were taken during machine milking (*n* = 6 for each diet group were randomly selected) for analysis. For more details on milking procedure and milk quality controls, see also Delgado-Pertíñez et al. [9]. The samples were frozen at −20 °C, or at −80 °C in the case of the samples for vitamins, until analysis was performed.

### 2.2. Chemical Analyses of Milk and Cheeses

The chemical composition (DM, protein, and fat percentages) of the milk was determined using near-infrared spectroscopy (NIRS), as described by Guzmán et al. [8]. The FA content (expressed as mg/g DM) was measured using the methods detailed by Guzmán et al. [8]. The fat was extracted from 0.1 g of freeze-dried milk or cheese, and the FA were directly methylated according to the procedures described by Sukhija and Palmquist [16] and Juárez et al. [17].

The retinol and α-tocopherol contents (expressed as μg/100 g) of 1.5–2 mL of milk or 2 g of cheese were estimated with the procedures of Herrero-Barbudo et al. [18] and Chauveau-Duriot et al. [19], as described by Guzmán et al. [8]. The total antioxidant capacity (TAC), expressed as μmol of water-soluble vitamin E analogue Trolox equivalents (TE), of 1 mL of milk or 2.5 g of cheese was determined by the ABTS (2,2′-azino-bis (3-ethylbenzothiazoline-6-sulphonic acid)) method of Fellegrini et al. [20], according to the procedure described by Gutiérrez-Peña et al. [21]. The total phenolic compounds (TPC), expressed as mg of gallic acid (GA) equivalents, of 8 mL of milk or 10 g of cheese were measured with the Folin–Ciocalteu method of Vázquez et al. [22] as described by Guzmán et al. [8] and Gutiérrez-Peña et al. [21].

### 2.3. Data Treatment and Statistical Analysis

The milk parameters were analysed with the one-way analysis of variance (ANOVA) model to test the dietary factor (CD, DOP40, or DOP80) as a fixed effect. The cheese composition data were subjected to a two-way factorial ANOVA analysis with dietary treatment and rennet (animal or vegetable) as fixed factors. Tukey’s honestly significant difference test was used for pairwise comparisons of the means. Finally, the Pearson correlation coefficients were also determined for some variables used in the analysis. In all analyses, a significance level of *p* ≤ 0.05 was considered statistically significant. The analyses were carried out using IBM SPSS Statistics v. 29.0 for Windows (IBM Corp., Armonk, NY, USA).

## 3. Results and Discussion

### 3.1. Antioxidant Compounds

The chemical characteristics of the milk samples taken from individual animals at the same time as the bulk milk collection for cheese manufacture are presented in Table 2. As previous studies in early lactation [8] and above the entire lactation period [2,9], the diets did not affect (*p* > 0.05) any proximal chemical parameters; however, the antioxidant parameters, except for retinol, were significantly affected by the diets. The α-tocopherol, TPC, and TAC contents were higher for the DOP80 diet than for the CD diet, while the DOP40 diet presented an intermediate value (*p* < 0.001).

The effects of the experimental factors (diet and type of rennet) on the antioxidant parameters of cheese are presented in Table 3. Except for retinol content, which was not modified by including DOP in the diet, α-tocopherol (*p* < 0.01), TPC (*p* < 0.001), and TAC (*p* < 0.001) contents were significantly affected by the diets. Specifically, the amounts in the cheeses increased as the amount of DOP replacing cereals increased. The α-tocopherol, TPC, and TAC contents were higher for the DOP80 diet (331.1 μg/100 g; 499.9 mg GA equivalents/kg; 95.1 μmol TE/g, respectively) than for the control diet (120.3 μg/100 g; 315.2 mg GA equivalents/kg; 49.9 μmol TE/g, respectively). For the DOP40 diet, these values were 229.2 μg/100 g, 430.7 mg GA equivalents/kg, and 69.0 μmol TE/g, presenting an intermediate value between the other two groups.

Moreover, there was a significant interaction between diet and type of rennet for TAC (*p* < 0.01; Table 3 and Figure 1). While the TAC values did not differ between the animal and vegetal rennet in the CD and DOP40 groups, the cheese from the DOP80 group and made with vegetable enzyme exhibited lower TAC values than the cheese made with animal rennet. Only a main effect for rennet was observed in the TPC content, with a higher value in cheeses made with animal rennet (493.2 mg GA equivalents/kg) than those made with vegetable coagulant (337.3 mg GA equivalents/kg). Finally, Figure 2 shows the positive correlations between TAC and some antioxidants, such as TPC (r = 0.73, *p* = 0.001) and α-tocopherol (r = 0.62, *p* = 0.008).

Antioxidants are included in the composition of dairy products such as vitamins, oligosaccharides, peptides, and minerals [11] and appreciable amounts of phenolic compounds [2,23,24]. The diet of the animal is one of the main points that influences the composition of the milk and cheese it produces, but in addition, bioactive compounds, such as vitamins and phenolic compounds, may be released in cheese through microbial metabolism [25,26]. Thus, the antioxidants from DOP in the current investigation were transferred from the feed to the milk and cheese, as with the results obtained in the previous studies in goats [2] and cows [27].

As the percentage of substitution of cereals by DOP in the diet increases, the contents of α-tocopherol and TPC in cheeses also increase, which would also explain the increase in TAC that agreed with the previous study by Delgado-Pertíñez et al. [2]. In addition to the bioactive peptides, of which cheese is considered a source [28], the high correlation obtained between TAC and the TPC and α-tocopherol contents (Figure 2) indicates the important role played by these compounds in improving the antioxidant capacity of the cheese. In agreement with other studies, the antioxidant activity of cheese was significantly correlated with the content of phenolic compounds [29,30] and α-tocopherol [31,32]. In the previous study [2], a correlation was only observed between TAC and TPC in milk, not with α-tocopherol. The correlation between TAC and α-tocopherol in cheese observed in the current study could be because cheese is a more concentrated product than milk. Compounds present in low concentrations in milk could be more concentrated and detected in cheese [33,34] and could provide evidence of the loss of milk components, such as proteins, minerals, fat, and lactose, into whey [35]. Unlike other research in cheese [36] and milk [37], retinol, a substance with antioxidant activity, had no significant correlation with TAC. This lack of effect is because neither the milk nor the cheese showed differences in this compound due to the diet and type of rennet.

Few studies have examined the effect of rennet substitutes on the total antioxidant activity in cheese [38,39,40], and no trials dealing with the effect on specific antioxidants, such as phenolic compounds and fat-soluble vitamins, have been reported. In the present study, only TPC and TAC were affected by the type of rennet used (Table 3). This effect could be due to differences in microbial metabolism [25,26] derived from differences in the enzymatic activity of each type of coagulant; however, this topic needs further investigation. Furthermore, the higher content of TPC in cheeses made with animal rennet could be explained, as mentioned above, by a concentration effect, probably as a consequence of the loss of milk components into the whey. These cheeses had a significantly higher percentage of DM (75%) than those made with a vegetable coagulant (72.6%), results obtained in the previous study by Guzmán et al. [13]. Considering the correlation between the TPC and TAC, the higher TPC content in these cheeses would also explain the higher TAC in the cheese made with animal rennet from goats fed with the DOP80 diet (Figure 1). In contrast to our results, Timón et al. [38] found no differences in the TACs of cow cheeses manufactured with animal and plant rennet, as evaluated by the DPPH method (2,2-diphenyl-1-picrylhydrazyl).

The findings outlined underscores the potential of DOP as a suitable alternative feed for improving the antioxidant status of the cheese produced, while also shed light on the potential benefits of incorporating agroindustrial by-products into animal diets to enhance the sustainability of livestock systems and contribute to the circular bioeconomy.

### 3.2. Fatty Acid Composition

The FA content of the milk is shown in Table 2. The samples were taken at the beginning of the fifth month of lactation (between the end of the mid-lactation stage and the beginning of the late lactation stage) and were not modified by diet for any of the variables analysed (*p* > 0.05) and agreed with the results found in the previous studies of the early [8] and mid-lactation phases [2]. Although significant interactions between the main factors (diets and lactation stage) were obtained for most FA in the same research and throughout lactation, the differences between diets were especially important in the late stage [2].

The content and indices of the FA of cheese according to diets and rennet are presented in Table 4. Except for the n-6:n-3 index, which was significantly lower (*p* < 0.05) for the DOP80 diet than for the DOP40 diet, none of the FA parameters were affected by the diet (*p* > 0.05). The FA composition of cheese depends on different factors, such as the technology used for cheese manufacturing and lipolytic activity during ripening [21,41,42,43], but also reflects the FA profile of the milk collected for cheese manufacture [34,44]. Thus, the results obtained in the current trial agree with the FA profile of the milk used to cheese-making.

No interactions were observed between the main factors, diet and rennet (Table 4). However, concerning the rennet applied, significant differences were obtained in the contents of 11 FA (Table 4): except for C15:1, all of them were higher in the cheeses made with animal rennet than in those made with vegetable rennet. The n-6:n-3 ratio was also significantly lower in the animal rennet cheeses. It has been reported that the FA were not changed by the coagulant used (calf rennet and plant coagulant) [45]. The results of the current work could be explained by differences in the enzymatic activity of each type of coagulant because vegetable enzymes, compared to animal coagulant, show slower lipolysis and faster proteolysis [14,46]. However, studies [47,48,49] show commercial liquid or powdered rennet, such as those used in the present study, would not present lipolytic activities, unlike rennet in the form of paste [50]. As for antioxidant compounds, the higher FA content obtained in cheeses made with animal coagulant could also be explained by a concentration effect due to its higher percentage of DM compared to cheeses manufactured with vegetable coagulant [13].

## 4. Conclusions

The DOP by-product is a suitable replacement for cereals in the diet of dairy goats and improves the antioxidant status of cheese for human health because the contents of α-tocopherol, TPC, and TAC of the cheeses increased as the percentage of DOP replacing cereals increased. Moreover, the high correlation obtained between the TAC and the TPC and α-tocopherol indicates the important role played by these compounds in improving the antioxidant activity of the cheese. Furthermore, replacing animal rennet with vegetable coagulant did not substantially influence the properties of artisanal cheeses made from raw milk.

## Figures and Tables

**Figure 1 vetsci-11-00171-f001:**
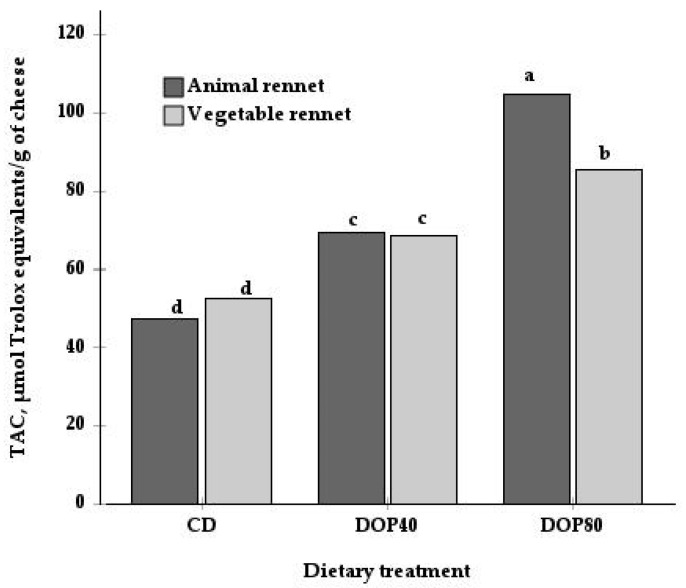
Interactive effect of the dietary treatment (DC, DOP40, and DOP80) and the type of rennet (animal and vegetable) on the total antioxidant capacity (TAC) of cheeses. Values presented are the means. Diets were: control diet (CD), consisting of alfalfa hay and a commercial concentrate; for the DOP40 and DOP80 diets, 40% and 80% of the cereal was replaced with dehydrated orange pulp (DOP). ^a,b,c,d^ Show differences between mean values (*p* < 0.05).

**Figure 2 vetsci-11-00171-f002:**
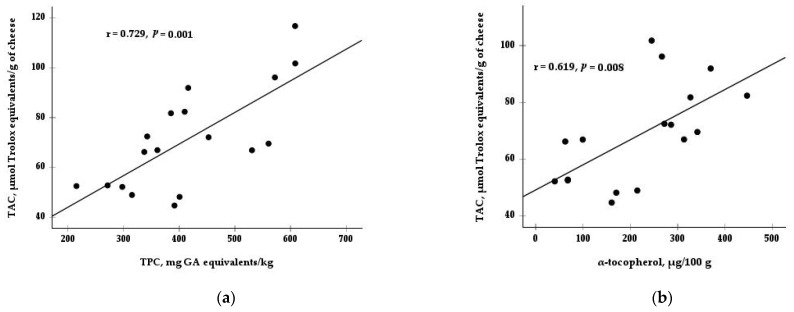
Correlation between total antioxidant capacity (TAC) and (**a**) total phenolic compounds (TPC) and (**b**) α-tocopherol in cheeses. GA = gallic acid.

**Table 1 vetsci-11-00171-t001:** Ingredients and chemical and nutritive composition of the different isoenergetic and isoproteic diets used to feed goats [13].

Items ^2^	Experimental Diets ^1^		
	CD	DOP40	DOP80
Ingredients, % dry matter (DM) basis			
Alfalfa hay	20.2	20.3	20.4
Concentrate			
Dehydrated orange pulp (pellets)	0.00	19.4	38.6
Oats	21.4	12.8	4.24
Barley	8.28	4.96	1.65
Corn	18.8	11.3	3.77
Soy flour, 44%	7.09	9.92	12.6
Sunflower pellets, 28%	12.5	12.1	13.5
Peas	10.0	7.87	3.93
Salt	0.39	0.39	0.39
Stabilised lard	0.39	0.00	0.00
Vitamins and minerals ^3^	1.01	1.01	1.02
Proximate composition and nutritive value, % DM			
DM, %	87.1	87.1	88.1
CP	20.9	18.7	18.3
NDF	29.8	26.6	28.3
ADF	14.7	15.2	16.8
ADL	3.09	3.13	3.43
EE	2.63	1.85	1.43
Ash	6.50	7.47	8.64
GE, kcal/g DM	4.37	4.31	4.25
UFL/kg	0.98	0.98	0.96
PDI	10.4	10.4	11.4

^1^ The control diet (CD) consisted of a commercial concentrate and alfalfa hay. For the DOP40 and DOP80 diets, 40% and 80% of the cereal was replaced with dehydrated orange pulp (DOP); ^2^ Nutral cabras LD granulado, Cargill^®^, Barcelona, Spain; ^3^ CP, crude protein; NDF, neutral detergent fibre; ADF, acid detergent fibre; ADL, acid detergent lignin; EE, ether extract; GE, gross energy; UFL, forage unit for lactation; PDI, protein digestible in the intestine.

**Table 2 vetsci-11-00171-t002:** Mean values of chemical characteristics of milk samples corresponding to experimental groups.

Item ^2^	Diet ^1^			SEM
	Control	DOP40	DOP80	
Dry matter (DM), %	11.7	12.3	12.6	0.16
Crude protein, %	3.17	3.18	3.34	0.06
Fat, %	3.50	4.09	4.15	0.13
Retinol, μg/100 g	2.82	5.98	6.15	0.64
α-Tocopherol, μg/100 g	17.9 ^b^	28.6 ^ab^	58.0 ^a^	5.88
TPC, mg GA equivalents/L	55.6 ^c^	72.8 ^b^	98.8 ^a^	4.48
TAC, μmol Trolox^®^ equivalents/mL	6.19 ^c^	8.95 ^b^	11.86 ^a^	0.61
Fatty acids (FAs), mg/g DM				
C4:0	7.99	9.33	8.30	0.51
C6:0	10.82	12.41	10.29	0.75
C8:0	9.37	10.65	8.77	0.65
C10:0	24.4	27.1	23.2	1.84
C12:0	12.6	14.3	11.8	0.88
C14:0	15.5	17.6	14.6	1.08
C14:1	0.42	0.48	0.40	0.03
C15:0	1.19	1.30	1.13	0.09
C16:0	54.8	62.6	50.7	3.90
C16:1	2.31	2.51	2.18	0.17
C17:0	0.73	0.84	0.69	0.05
C17:1	0.20	0.23	0.19	0.01
C18:0	24.8	27.0	23.4	1.79
C18:1 n-9 *trans*	2.17	2.46	2.03	0.15
C18:1 n-11 *trans* (VA)	1.54	1.75	1.44	0.11
C18:*1* n-9 *cis*	45.2	49.2	42.6	3.27
C18:2 n-6 *trans*	0.33	0.38	0.31	0.02
C18:2 n-6 *cis*	7.57	8.60	7.08	0.53
α-C18:3 n-3	0.48	0.55	0.45	0.03
γ-C18:3 n-6	0.20	0.26	0.19	0.02
CLA *cis-9*, *trans-11* (RA)	1.58	1.72	1.49	0.11
C20:0	0.48	0.55	0.45	0.03
C20:4 n-6 (ARA)	0.44	0.50	0.41	0.03
C20:5 n-3 (EPA)	0.06	0.07	0.06	0.00
C22:0	0.26	0.30	0.25	0.02
C22:5 n-3 (DPA)	0.13	0.14	0.12	0.01
C22:6 n-3 (DHA)	0.05	0.06	0.05	0.00
Others	1.00	1.14	0.95	0.01
SFA	163.4	184.4	154.0	11.5
MUFA	52.1	56.9	49.1	3.75
PUFA	11.2	12.6	10.5	0.78
Total CLA	1.61	1.75	1.52	0.12
n-6	8.61	9.82	8.08	0.60
n-3	0.78	0.88	0.74	0.06
n-6:n-3	11.0	11.2	11.0	0.06

Means with different letters (a, b, c) within each row differ significantly (*p* ≤ 0.05). ^1^ The control diet consisted of alfalfa hay and a commercial concentrate. For the DOP40 and DOP80 diets, 40% and 80% of the cereal was replaced with dehydrated orange pulp (DOP); ^2^ TPC, total phenolic compounds; GA, gallic acid; TAC, total antioxidant capacity; VA, vaccenic acid; RA, rumenic acid; ARA, arachidonic acid; EPA, eicosapentaenoic acid; DPA, docosapentaenoic acid; DHA, docosahexaenoic acid; SFA, saturated FAs; MUFA, monounsaturated FA; PUFA, polyunsaturated FAs; CLA, conjugated linoleic acid.

**Table 3 vetsci-11-00171-t003:** Mean values of fat-soluble vitamins, phenolic compounds, and total antioxidant capacity of cheeses according to experimental diets and rennet.

Item ^2^	Diet ^1^ (D)			Rennet (R)		SEM	*p*-Values		
	Control	DOP40	DOP80	Animal	Vegetable		D	R	D × R
Retinol, μg/100 g	26.9	26.9	23.2	22.8	28.6	2.62	0.803	0.291	0.209
α-Tocopherol, μg/100 g	120.3 ^b^	229.2 ^ab^	331.1 ^a^	223.1	218.6	29.9	0.009	0.848	0.097
TPC, mg GA equivalents/kg	315.2 ^c^	430.7 ^b^	499.9 ^a^	493.2 ^a^	337.3 ^b^	27.7	0.000	0.000	0.158
TAC, μmol Trolox^®^ equivalents/g	49.9 ^c^	69.0 ^b^	95.1 ^a^	73.9	68.8	4.83	0.000	0.065	0.004

Means with different letters (a, b, c) within each row and factor differ significantly (*p* ≤ 0.05); ^1^ The control diet consisted of alfalfa hay and a commercial concentrate. For the DOP40 and DOP80 diets, 40% and 80% of the cereal was replaced with dehydrated orange pulp (DOP); ^2^ TPC, total phenolic compounds; GA, gallic acid; TAC, total antioxidant capacity.

**Table 4 vetsci-11-00171-t004:** Mean values of the fatty acid composition of cheeses according to the experimental diets and rennet.

Fatty Acids ^2^, mg/g DM	Diet ^1^ (D)			Rennet (R)		SEM	*p*-Values		
	Control	DOP40	DOP80	Animal	Vegetable		D	R	D × R
C4:0	17.0	17.1	16.9	17.8	16.2	0.75	0.996	0.366	0.496
C6:0	25.3	21.9	21.4	23.6	22.1	1.05	0.327	0.527	0.777
C8:0	22.1	18.9	19.6	20.8	19.6	0.96	0.418	0.568	0.676
C10:0	89.8	79.5	86.4	93.2	77.3	5.01	0.700	0.141	0.450
C11:0	1.08	0.98	1.23	1.06	1.14	0.06	0.259	0.507	0.120
C12:0	48.4	43.6	53.3	51.1	45.7	2.83	0.409	0.363	0.446
C13:0	0.87	0.94	0.97	1.05 ^a^	0.80 ^b^	0.06	0.769	0.048	0.923
C14:0	88.1	78.2	82.7	89.8	76.2	4.92	0.724	0.194	0.360
C14:1	2.49	1.96	2.29	2.73 ^a^	1.76 ^b^	0.19	0.364	0.006	0.231
C15:0	5.53	4.70	5.89	5.63	5.12	0.31	0.328	0.438	0.619
C15:1	1.43	1.42	1.78	1.30 ^b^	1.79 ^a^	0.10	0.123	0.006	0.199
C16:0	250.2	227.2	236.6	262.5	213.5	13.9	0.793	0.100	0.439
C16:1	11.4	10.8	11.9	10.0	12.8	0.68	0.789	0.058	0.583
C17:0	3.69	3.64	3.32	3.21	3.89	0.22	0.773	0.158	0.577
C17:1	1.93 ^b^	2.33 ^ab^	2.86 ^a^	2.65	2.10	0.17	0.048	0.067	0.299
C18:0	62.1	53.7	57.8	65.4 ^a^	50.4 ^b^	3.67	0.625	0.050	0.462
C18:1 n-9 *trans*	3.35	2.90	3.86	3.04	3.71	0.20	0.127	0.083	0.710
C18:1 n-11 *trans* (VA)	3.38	2.82	3.63	4.02 ^a^	2.54 ^b^	0.30	0.337	0.006	0.073
C18:*1* n-9 *cis*	185.8	152.5	149.5	184.5	140.7	11.2	0.318	0.056	0.663
C18:2 n-6 *trans*	3.99	3.57	3.89	4.60 ^a^	3.02 ^b^	0.29	0.709	0.003	0.087
C18:2 n-6 *cis*	19.5	20.7	21.3	21.1	19.9	1.03	0.813	0.620	0.559
α-C18:3 n-3	2.61	2.72	3.41	3.30 ^a^	2.52 ^b^	0.20	0.148	0.038	0.262
γ-C18:3 n-6	0.93	0.84	0.89	0.96	0.82	0.06	0.850	0.346	0.461
CLA *cis-9*, *trans-11* (RA)	5.22	4.42	4.38	4.83	4.51	0.27	0.442	0.590	0.713
CLA n-10 *trans*, n-12 *cis*	0.46	0.33	0.35	0.45 ^a^	0.31 ^b^	0.03	0.093	0.009	0.156
C20:0	0.41	0.30	0.31	0.39	0.29	0.03	0.095	0.055	0.349
C20:1 n-9	0.27	0.23	0.24	0.28	0.22	0.02	0.538	0.098	0.519
C20:2	0.32	0.32	0.30	0.34	0.29	0.02	0.923	0.219	0.833
C20:3 n-3	0.99	0.83	0.84	0.99	0.78	0.06	0.447	0.092	0.282
C20:3 n-6	0.41	0.36	0.37	0.41	0.35	0.02	0.716	0.274	0.306
C20:4 n-6 (ARA)	4.22	4.14	4.80	4.87	3.90	0.27	0.523	0.076	0.302
C20:5 n-3 (EPA)	1.06	0.99	1.04	1.16	0.91	0.06	0.903	0.056	0.259
C21:0	0.33	0.27	0.27	0.34 ^a^	0.24 ^b^	0.02	0.247	0.011	0.251
C22:0	0.92	0.72	0.81	0.92	0.72	0.06	0.328	0.074	0.256
C22:1 n-9	0.18	0.16	0.19	0.20	0.15	0.01	0.469	0.055	0.475
C22:2	0.09	0.10	0.10	0.11	0.09	0.01	0.499	0.126	0.478
C22:5 n-3 (DPA)	1.03	0.99	1.04	1.15 ^a^	0.89 ^b^	0.06	0.931	0.045	0.512
C22:6 n-3 (DHA)	0.91	0.87	0.93	0.97	0.83	0.05	0.864	0.202	0.466
C23:0	0.19	0.17	0.17	0.20 ^a^	0.15 ^b^	0.01	0.796	0.042	0.946
C24:0	0.26	0.25	0.26	0.27	0.24	0.01	0.925	0.273	0.498
C24:1	0.18	0.15	0.19	0.19	0.16	0.01	0.260	0.091	0.523
SFA	616.4	552.1	588.0	637.3	533.7	33.2	0.740	0.148	0.465
MUFA	210.3	175.4	176.5	208.9	165.9	12.1	0.408	0.091	0.636
PUFA	41.7	41.1	43.7	45.2	39.1	2.25	0.903	0.227	0.527
CLA total	5.67	4.75	4.73	5.28	4.82	0.23	0.402	0.476	0.666
n-6	29.1	29.6	31.3	31.9	28.0	1.57	0.855	0.265	0.527
n-3	6.59	6.40	7.27	7.58	5.93	0.42	0.643	0.057	0.353
n-6:n-3	4.46 ^ab^	4.62 ^a^	4.37 ^b^	4.24 ^b^	4.72 ^a^	0.07	0.037	0.000	0.100

Means with different letters (a, b) within each row and factor differ significantly (*p* ≤ 0.05); ^1^ The control diet consisted of alfalfa hay and a commercial concentrate. For the DOP40 and DOP80 diets, 40% and 80% of the cereal was replaced with dehydrated orange pulp (DOP); ^2^ VA, vaccenic acid; RA, rumenic acid; ARA, arachidonic acid; EPA, eicosapentaenoic acid; DPA, docosapentaenoic acid; DHA, docosahexaenoic acid; SFA, saturated FAs; MUFA, monounsaturated FAs; PUFA, polyunsaturated FAs; CLA, conjugated linoleic acid.

## Data Availability

The data presented in this study are available on request from the corresponding author.

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
