# Peer review of "Feeding Dairy Goats Dehydrated Orange Pulp Improves Cheese Antioxidant Content"

_vetsci, 2024, doi:10.3390/vetsci11040171_

Round 1

Reviewer 1 Report

Comments and Suggestions for Authors

I reviewed the manuscript Number vetsci-2934451

entitled: Feeding Dairy Goats Dehydrated Orange Pulp Improves Cheese Antioxidant 
Content. 

Strengths: 

-       The study describes an interesting subject, investigating the effects of replacing cereal (0%, 40% and 80%) with dehydrated orange pulp in the diet of Payoya dairy goats on the antioxidant compounds and fatty acid contents of artisanal cheeses made from raw milk and presenting some valuable results.

-       The manuscript is well-written and reports interesting results, the information presented were new regarding cheese properties and the conclusions supported by the data.

-       The methodology used was very clear, allowing the reader to follow the results easily.

General comments: It would be very interesting to also present the economical benefits from the use of dehydrated orange pulp instead of cereal in the diet of goats.

Reviewer 2 Report

Comments and Suggestions for Authors

Dear authors of "Feeding Dairy Goats Dehydrated Orange Pulp Improves Cheese Antioxidant Content",

Your research on the effects of replacing cereal with dehydrated orange pulp (DOP) in the diet of goats on the antioxidant and fatty acid contents of cheeses is both innovative and significant. The findings outlined in your abstract shed light on the potential of agroindustrial by-products, such as DOP, to enhance the sustainability of livestock systems and contribute to the circular bioeconomy.

The analysis conducted on the milk-clotting process with both animal and vegetable rennet provides valuable insights into the role of coagulant enzymes in determining cheese properties. Moreover, the observed increase in α-tocopherol levels, total phenolic compounds (TPC), and total antioxidant capacity (TAC) in cheeses as the percentage of DOP replacing cereals increases underscores the potential of DOP as a suitable alternative feed for goats, while also improving the antioxidant status of the cheese produced.

Your study contributes significantly to our understanding of sustainable livestock practices and the potential benefits of incorporating agroindustrial by-products into animal diets. We believe that your findings will be of great interest to our readership and will make a valuable addition to the scientific literature in this field.

Despite of this, I have detected some aspects to improve

-the simple summary does not cover the requirements of the journal, it seems a mere repetition of the abstract

-there is an inadequate number of self-citations, there are more studies on this subject made by other authors that can be added in the introduction or in the discussion as.

Reviewer 3 Report

Comments and Suggestions for Authors

Lines 423 and 483: put scientific names in italics. Sulla coronarium and Cynara cardunculus

Reviewer 4 Report

Comments and Suggestions for Authors

The area of sustainable agrifood& circular economy are  very actual topics. The Authors were testing fruit by- product in animal feed (goats)& the effect of the orange pulp on parameters of cheese made from the raw milk.

Abstract gives suitable information.

Authors tested by-product which is highly produced in Spain - so for the future perspective the supply of DOP would be easy. Spain is also big producers of goat cheese. And if the idea will work, would have serious impact on local economy.Authors have already published papers from the research on DOP in feed. So the novelty of the current study in comparison to previous one is focusing on DOP in the diet of dairy goats effects on the antioxidant compounds and fatty acid (FA) contents  of cheese.

The diet composition, animals & cheese making were described in previous paper. 

Number of animals used in the experiment: 44, divided into 3 groups. 

Table 1 - details of chemical analysis

In the methods part there is lack od details about milk parameters/controls/quality/safety.

milking technique?

 any sensory analysis of cheeses?

line 254 its hard to evaluate microbial metabolism when microbiology wasnt checked for this study - or is not presented in this paper.

more data about rennet functionalities would be valuable

references list is actual & fits to subject

for the reviewer the discussion is quite "flat" lacks in dynamics. The Authors shuld put more emphasis on the novelty of the study.

no serious faults in the manuscript, the reviewer sees the needs of addition of some data to the M&M part

Reviewer 5 Report

Comments and Suggestions for Authors

Very well written MS.  Only criticism is that they do reference their research group extensively, but there may not be significant other work done besides theirs on the subject.
